# *Clostridium**n**ovyi’s* Alpha-Toxin Changes Proteome and Phosphoproteome of HEp-2 Cells

**DOI:** 10.3390/ijms23179939

**Published:** 2022-09-01

**Authors:** Theresa Schweitzer, Harald Genth, Andreas Pich

**Affiliations:** 1Institute of Toxicology, Hannover Medical School, Carl-Neuberg-Str. 1, 30625 Hannover, Germany; 2Core Facility Proteomics, Hannover Medical School, Carl-Neuberg-Str. 1, 30625 Hannover, Germany

**Keywords:** *C. novyi*, *Clostridium novyi*, TcnA, LCGTs, *C. difficile*, TcdA, TcdB, proteomics, pathway analysis, phosphoproteomics

## Abstract

*C. novyi* type A produces the alpha-toxin (TcnA) that belongs to the large clostridial glucosylating toxins (LCGTs) and is able to modify small GTPases by N-acetylglucosamination on conserved threonine residues. In contrast, other LCGTs including *Clostridioides difficile* toxin A and toxin B (TcdA; TcdB) modify small GTPases by mono-o-glucosylation. Both modifications inactivate the GTPases and cause strong effects on GTPase-dependent signal transduction pathways and the consequent reorganization of the actin cytoskeleton leading to cell rounding and finally cell death. However, the effect of TcnA on target cells is largely unexplored. Therefore, we performed a comprehensive screening approach of TcnA treated HEp-2 cells and analyzed their proteome and their phosphoproteome using LC-MS-based methods. With this data-dependent acquisition (DDA) approach, 5086 proteins and 9427 phosphosites could be identified and quantified. Of these, 35 proteins were found to be significantly altered after toxin treatment, and 1832 phosphosites were responsive to TcnA treatment. By analyzing the TcnA-induced proteomic effects of HEp-2 cells, 23 common signaling pathways were identified to be altered, including *Actin Cytoskeleton Signaling*, *Epithelial Adherens Junction Signaling*, and *Signaling by Rho Family GTPases*. All these pathways are also regulated after application of TcdA or TcdB of *C. difficile*. After TcnA treatment the regulation on phosphorylation level was much stronger compared to the proteome level, in terms of both strength of regulation and the number of regulated phosphosites. Interestingly, various signaling pathways such as *Signaling by Rho Family GTPases* or *Integrin Signaling* were activated on proteome level while being inhibited on phosphorylation level or vice versa as observed for the *Role of BRCA1 in DNA Damage Response*. ZIP kinase, as well as Calmodulin-dependent protein kinases IV & II, were observed as activated while Aurora-A kinase and CDK kinases tended to be inhibited in cells treated with TcnA based on their substrate regulation pattern.

## 1. Introduction

*Clostridium novyi* is an obligate anaerobic pathogen that causes serious disease in both animals and humans. It is the main cause of gas gangrene in humans, while it is involved in infectious necrotic hepatitis in animals [1,2]. The exact identification of *C. novyi* and its contribution to the course of both diseases is often difficult due to mixed clostridial populations of *Clostridium perfringens*, *C. novyi*, *Clostridium septicum*, *Clostridium hystolyticum*, or others causing these diseases [1]. Pathogenic *C. novyi* strains are classified by different toxin productions. Type A strains produce alpha, gamma, delta, and epsilon toxins [3,4,5]. Type B produces alpha, beta, and zeta toxins, and *C. haemolyticum*, also known as *C. novyi* type C is capable of producing beta, eta, and theta toxins [3,4,5,6,7]. *C. novyi* type D does not produce any toxin and is generally referred to as nonvirulent.

Alpha Toxin (TcnA) is one of the major virulence factors of *C. novyi* and belongs to the family of large clostridial glucosylating toxins (LCGTs) [8]. Other LCGTs are the *C. difficile* toxins A and B (TcdA, TcdB), the lethal and hemorrhagic toxin of *Clostridium sordellii* (TcsH, TcsL), and the large cytotoxin of *C. perfringens* (TpeL) [9]. LCGTs exhibit a four-domain structure. The receptor-binding domain is located at the c-terminus and mediates the initial binding of the toxin to the cell surface. Receptor-mediated endocytosis follows, in which the translocation domain, which is adjacent to the receptor-binding domain, enables toxin translocation through the membrane of the acidified endosome into the cytosol. The N-terminal glucosyltransferase domain is cleaved off by the cysteine protease domain and exerts its enzymatic activity in the cytosol [10,11].

LCGTs inactivate small GTPases by glycosylating conserved threonine residues (Thr-35 in Rac/Thr-37 in RhoA) [12]. Most LCGTs exploit UDP-glucose as the sugar donor for this process, which leads to mono-O-glucosylation. However, TcnA preferably uses UDP-N-acetylglucosamine (UDP-GlcNac), which leads to Mono-O-N-acetylglucosamination of the GTPases [12,13,14]. Crucial for co-substrate specificity of LCGTs are the amino acids at positions 383 and 385 in the catalytic cleft. For UDP-glucose favoring toxins such as TcdB, those positions contain Ile-383 and Glu-385 while TcnA favoring UDP-GlcNac comprises Ser-383 and Ala-385. Isoleucine and glutamic acid contain bulkier side chains than serine and alanine, respectively, thus limiting the available space for the co-substrate [15]. By alteration of GTPase signaling, there is also an impact on the actin cytoskeleton organization [16]. Data on the consequences of TcnA effects on GTPases are hardly available while in contrast, TcdA and TcdB have been extensively studied.

Herein we describe the effects of TcnA on an epithelial cell line called HEp-2. It has been established via HeLa cell contamination and expresses LDLR as well as SLC35B2, which are key factors in initial cellular binding and uptake of TcnA [17]. We performed a global DDA proteome analysis [18] to screen for TcnA responsive proteins and phosphosites after treatment with TcnA. The phosphoproteome responds much faster to altered conditions, e.g., addition of TncA, and should lead to the identification of important regulatory proteins. Phosphorylation is a post-translational modification that proceeds much faster than the synthesis or degradation of mature proteins. With the reproducible enrichment and purification of phosphoproteins and peptides using metal affinity strategies, it has become possible to analyze the phosphoproteome in a comprehensive way [19,20]. Additionally, we compared the impact of TcnA on the proteome and phosphoproteome of target cells with the corresponding effects of TcdA and TcdB [21,22].

## 2. Results

### 2.1. Morphological Alterations of HEp-2 Cells Treated with C. novyi’s Alpha Toxin

Morphological changes of HEp-2 cells were analyzed using phase-contrast microscopy (Figure 1). Three different treatments were compared: no toxin (Ctrl), TcnA inactivated with formaldehyde (FA_TcnA) and active TcnA (TcnA). TcnA concentration of 255 ng/mL were used. First morphological changes were visible after 3–4 h (data not shown) and after 24 h most of the cells were completely round (Figure 1), except those treated with no toxin and FA_TcnA. Along with cell rounding, cell adhesion to the flask was reduced and led to cell strands ranging into the medium, indicated by white arrows in Figure 1. Increasing TcnA concentrations or incubation times did not lead to a higher percentage of completely rounded cells without being accompanied by undesirable side effects such as apoptotic or necrotic cells (data not shown). Cell morphology of FA_TcnA looked similar to control cells.

### 2.2. Generation of a Control Toxin to Elucidate TcnA Effects on Proteome and Phosphoproteome

To detect the effects of TcnA on cellular protein homeostasis using MS-based proteomics, an appropriate control needs to be used. Since no catalytically inactive mutant was available for this purpose (as for TcdA and TcdB), the toxin was inactivated with formaldehyde and subsequently tested on HEp-2 cells. Effects of this inactivated toxin were analyzed on proteome and phosphoproteome level in comparison to the active toxin and non-treated control cells.

For the comparison on proteome level, data from 5086 identified and quantified protein groups were used that have been obtained from the three treatment conditions TcnA, FA_TcnA, and control (proteome data see Appendix A). In a *Benjamini Hochberg* (BJH; FDR = 0.05) [23] corrected analysis of variance (ANOVA) test 530 proteins showed significant differences in their abundance across the three treatments. Hierarchical clustering with BJH positive proteins was performed after z-scoring (Figure 2A). FA_TcnA and Ctrl samples showed similar regulated protein clusters, whereas a significant difference was observable in comparison to the TcnA treated cells. Only one protein (Zinc Finger Protein 611, Znf611, see Appendix A) was detected as significantly down-regulated between control and FA_TcnA treatment, showing that TcnA inactivation by formaldehyde was successful and the influence of potentially co-enriched other factors during toxin purification on HEp-2 cells should be unlikely. Znf611 was excluded from further proteome analysis of protein regulation between TcnA and FA_TcnA treated replicates. Further analyses of detailed TcnA-induced alterations concerning the proteome are presented below.

To further control the suitability of the FA-inactivated TcnA a comparison of phosphoproteome was performed. Overall, 9427 phosphosites belonging to 2125 phosphoproteins that had been normalized to the corresponding protein abundance determined in the proteome measurements were used. This way, direct comparison of active and inactivated TcnA was possible (phosphoproteome data see Appendix A).

*BJH* (FDR = 0.05) corrected ANOVA testing identified 430 phosphosites as significantly regulated across the three treatments. Hierarchical clustering with *BJH*-positive phosphosites revealed that FA_TcnA-treated cells and control cells shared more similarities than TcnA treated cells (Figure 2B). However, within FA_TcnA treated cells eight phosphosites showed a significant down-regulation while four phosphosites were significantly up-regulated (Appendix A). These twelve phosphosites were excluded from further bioinformatic analysis of regulated phosphosites in TcnA versus FA_TcnA treated cells. Overall, most of the alterations induced by FA_TcnA clustered with control conditions, both on proteome and phosphoproteome level.

Thus, the formaldehyde inactivated TcnA has been established as a valuable control to study the effects of TcnA on HEp-2 cells.

### 2.3. Proteomic Effects Induced by TcnA in HEp-2 Cells

The impact of TcnA on target cells was compared to FA_TcnA, which was used as control. The comprehensive proteome analysis was based on 5086 identified and quantified protein groups using a TMT approach (Appendix A). In TcnA-treated cells, 14 proteins were significantly up-regulated, while 21 proteins were significantly down-regulated (Figure 3A). The log_2_ expression differences ranged from −2.34 to 2.58 between treated and untreated samples. Among the strongest up-regulated proteins were Legumain (Lgmn) and Immunoglobulin-like and fibronectin type III domain-containing protein 1 (Igfn1), while proteins like Troponin I (Tnni2) as well as ATP-binding cassette sub-family D member 2 (Abcd2) were identified to be strongly down-regulated.

A BJH (FDR = 0.05) corrected t-test identified 1021 positive proteins which were used for Fisher’s exact test to identify TcnA-responsive gene ontology (GO) groups (Figure 3B). GO annotations based on three main classes, cellular component (GOCC), molecular function (GOMF) and biological process (GOBP) were the basis for this Fisher’s exact test. There are various subterms within the three GO main classes, which were annotated to the proteins, based on their localization, known biological or molecular functions. With the Fisher’s exact test, a comparison between the frequency of these subterms within the whole proteome and across the significantly regulated proteins was made. Most of the significantly enriched terms within the subset belonged to GOCC. The only identified GOMF term was *cytoskeletal protein binding*, while no terms concerning biological processes were enriched. It has to be mentioned that the annotation follows a hierarchical order. Therefore, all proteins with the annotation *actin cytoskeleton* were also annotated as *cytoskeleton*. The same applies to the term *actin filament bundle* which was always accompanied by the terms *stress fibers* and *actomyosin* being the least stringent term.

For canonical pathway analysis via IPA^TM^ all changed proteins with a significant *p*-value < 0.05 were used. IPA^TM^ uses *p*-value and z-score as statistical parameters to describe the quality of the predicted events. The *p*-value includes the amount of genes participating in a pathway (provided by IPA^TM^) and the overlap of those genes within the uploaded data. With a *p*-value < 0.05 a non-random association between pathway and data is present [24]. The z-score additionally describes the observed regulation status. A z-score below zero indicates inactivation of the pathway. The lower the value, the better the measured protein regulations agree with the inactivation of the pathway. A z-score above 0 shows activation of the pathway and, comprehensibly, the higher the value, the better the measured protein regulations agree with the activation of the pathway [25]. Some of the most significantly regulated pathways are illustrated (Figure 3C). The strongest activated pathways in TcnA treated cells were *Signaling by Rho Family GTPases* and *RHOA Signaling*, which contained a big overlap of proteins included in both. Of the significantly regulated proteins 37 were identified as part of *RhoA signaling* pathway, while 57 proteins were assigned to *Rho Family GTPases* pathway. An overlap of 29 proteins was present in both. The pathways *RHOGDI Signaling*, *Role of BRCA1 in DNA Damage Response* and *IGF-1 Signaling* were inhibited, while every other was predicted as at least slightly activated according to the z-score.

Comparison of IPA^TM^ canonical pathways altered by TcdA [21], TcdB [22], and TcnA in Hep-2 cells was performed in order to identify co- and differentially regulated cellular processes. It has to been mentioned that TcdA and TcdB analyses were done by stable isotope labeling with amino acids in cell culture (SILAC) approaches. Those were characterized by high reproducibility but lower degrees of protein regulation as well as overall lower numbers of quantifiable proteins than the TcnA TMT approach. This is why there were only 169 proteins passing significance level of *p* < 0.05 for TcdA and 168 proteins for TcdB while the TcnA analysis resulted in 2064 proteins with *p* < 0.05 used for the IPA analysis. For TcdA 39 regulated (z-score > 0; <0; *p* < 0.05) pathways could be identified. In average, the pathways tended to be inactivated after TcdA treatment. In contrast, for TcdB 118 regulated pathways were observed, most of which were activated. Following TcnA treatment, 162 regulated pathways were identified. In Figure 4, some of the pathways regulated by every LCGT treatment are depicted (for regulated pathways including z-scores see Table 1, all identified pathways are listed in Appendix A). The overlap between TcdB and TcnA treated cells is highest in IPA canonical pathway analysis, which is due to the fact that only 39 regulated pathways were identified with TcdA treatment. However, looking directly on the proteome level the overlap between TcdA and TcdB treatment is higher. A comparison of the top 20 regulated proteins per treatment is shown in Table 2. The proteins RhoB and Hmox1 showed pronounced up-regulation within TcdA and TcdB treatment, while Yap1 and Tpm2 are both down regulated. No overlap between any of the strongest regulated proteins after TcnA treatment to treatment with one of the other toxins was observable.

### 2.4. Phosphoproteomic Effects of TcnA on HEp-2 Cells

Phosphoproteomic analysis of TcnA treated HEp-2 cells revealed overall, 9427 identified phosphosites (Appendix A) with log_2_ difference between TcnA and FA_TcnA treatment ranging from −2.92 to 2.97. Out of these 161 phosphosites were identified as significantly down-regulated while 69 phosphosites were significantly up-regulated (Figure 5A). The range and amount of regulated phosphosites both reflect a higher degree of regulation than on protein level. Strongest up-regulated phosphosites were, e.g., Thr-124 of Serine/Threonine Protein Kinase N2 (Pkn2) and the Thr-594 of Mitogen-activated protein kinase 7 (Mapk7). Phosphosites Ser-313 of Plakophilin 3 (Pkp3) and Thr-343 of NK-tumor recognition protein (Nktr) showed pronounced down-regulation.

For phosphorylation analysis via the IPA^TM^ software all 1832 significantly different (*p* < 0.05) phosphosites were uploaded and 889 IPA entries could be matched. Twelve of the pathways with the highest regulation status are shown in Figure 5B. What has to be mentioned in the context of phosphorylation analysis with IPA is that every phosphosite is attributed as ‘activating the pathway’ so in this case the resulting information would not be ‘activated or inhibited pathway’ based on the z-score, but rather pathways which exhibit net-positive or net-negative phosphosite regulations. In comparison to the proteome data (Figure 3C) more pathways showed decreased phosphorylation events which goes along with more phosphosites being observed as downregulated in TcnA treated cells. *Actin Cytoskeleton* and *RHOA Signaling* showed upregulation on proteome level as well as on phosphorylation levels. *Signaling by Rho Family GTPases*, *Regulation of Actin-based Motility by Rho*, *Integrin* and *Insulin Receptor Signaling* showed decreased phosphorylation, while being up-regulated on protein level. *Rac-*, *Paxillin* and *Epithelial Adherens Junction Signaling,* which did not belong to the strongest regulated pathways on proteome level, also showed pronounced down-regulation of phosphosites. *Role of BRCA1 in DNA Damage Response* was identified as the pathway with the strongest, significant up-regulation of phosphosites (while being inhibited on proteome level). As a subnetwork of *Role of BRCA1 in DNA Damage Response*, *ATM Signaling* also exhibited pronounced up-regulation of attributed phosphosites, however with a lower z-score.

A noteworthy protein analyzed in phosphoprotein analysis was Serine/arginine repetitive matrix protein 2 (Srrm2), a nearly 300 kDa protein which is part of the spliceosome and exhibited 71 significant phosphosites.

By application of known kinase–substrate interactions to the collected phosphorylation data, predictions about the regulation of kinase activities are possible. In order to do this a Fisher’s exact test (*p*-value < 0.05) was performed based on the presence of kinase–substrate motif annotations. The log_2_ mean regulation of substrates (*p* < 0.05) were calculated for each kinase in order to predict their activation status. This bioinformatic analysis highlighted ZIP kinase as well as Calmodulin-dependent protein kinase IV and II (CaMK VI; CaMK II) as activated kinases with mean log_2_ ratios of 0.28; 0.25 and 0.07, respectively (Figure 5). Especially cyclin-dependent kinases showed predicted inactivation with CDK5 and Cdc2 (also known as CDK1) showing slightly more pronounced downregulation of substrates in comparison with the overall CDK kinase motif (log_2_ ratios −0.39; −0.27, −0.27). The protein kinases with strongest inactivation however are Aurora-A kinase as well as AMP-activated protein kinase (log_2_ ratio −0.42; −0.41). Overall kinase prediction identified most kinases as inactivated, with only ZIP kinase, Calmodulin-dependent protein kinase IV and II and Chk1 kinase eliciting increased phosphorylated substrates after TcnA treatment.

## 3. Discussion

For the first time a global proteome and phosphoproteome analysis of TcnA treated target cells was performed. HEp-2 cells were used as target cells since they have proven before to be sensitive to various LCGTs and express required key proteins for cellular uptake of TcnA [17,24,25]. In this way, it was possible to identify cellular pathways regulated through toxin administration and deepen the understanding of particular effects of TcnA and general effects of LCGTs on target cells.

First, we tried to generate a suitable control for TcnA-induced cellular effects. An inactive toxin variant was not available as it is the case for *Clostridioides difficile* toxins TcdA and TcdB, for which genetically engineered mutants had been generated with impaired glucosyltransferase activity [26,27]. Thus, we inactivated the purified TcnA by treatment with formaldehyde and named it FA_TcnA. The quality of this inactivated version was analyzed by comparing it to non-treated control cells on three different levels: (1) general morphology of treated cells observable via light-microscopy; (2) alterations on proteome level; and (3) by alteration on phosphoproteome level. Cell morphology was similar for cells treated with FA_TcnA and non-treated control cells. On proteome level only the zinc-finger protein Znf611 was affected by FA_TcnA treated cells and from 9500 quantified phosphosites only 12 were regulated between FA_TcnA treatment and control. This comparison has been performed in order to exclude effects of high or low molecular weight compounds that might have been co-purified during toxin purification. Overall, we can say that all three tests proved FA_TcnA to be a very well suited control to characterize TcnA effects on HEp-2 cells.

Following, the focus was set on observed proteomic alterations induced by TcnA. A total of 5086 proteins were identified and quantified in every replicate (Appendix A). Of those proteins, 14 expressed significant up-regulation and 21 significant down-regulation in response to TcnA (Figure 3). Two of those strongly regulated proteins (Tnni2 and Lgmn) and the interpretation of the data on single protein level will be discussed hereinafter.

The protein Legumain (Lgmn) showed up-regulation of factor 2.4 with strongest significance (−log_2_
*p*-value: 4.5). It is also referred to as asparagine endopeptidase (AEP) and primarily hydrolyzes asparaginyl bonds as well as aspartyl bonds at low pH, thus contributing to protein degradation [28,29,30]. It is mainly located within lysosomes but has also been detected extracellularly, in the nucleus or in the cytosol [31,32,33,34,35]. Each compartment with different pH influences its enzymatic functions [36]. In lysosomes, it takes part in antigen processing, while a carboxypeptidase function has been described at neutral pH and ligase activities have also been observed [37,38,39]. Since whole cell lysates were generated in our proteome analysis, it is not possible to tell whether Lgmn was enriched in a certain compartment, making hypothesis according to its function in cellular responses to TcnA tenuous. Probably Lgmn might be induced due to an increased protein turnover rate. However, to further specify the role of Lgmn in TcnA treated cells future experiments need to be carried out.

The protein Troponin I, fast skeletal muscle (Tnni2) is one of the strongest down-regulated proteins among the TcnA responsive proteins with a regulation factor of −4.7 and a *p*-value of 0.03. It is part of the inhibitory unit of the troponin complex which is mostly associated with striated muscles (skeletal or cardiac) and regulates muscle contraction [40]. The troponin complex is a combination of the Ca^2+^ binding protein TnC that connects the inhibitory subunit (TnI) and the tropomyosin binding subunit (TnT) [41]. By binding of Ca^2+^ a conformational change is induced which leads to exposition of myosin binding sides on the actin strands [42]. However, while the presence of Tnni2 in various epithelial cells has been reported, its exact function remains poorly explored. Sawaki et al. observed Tnni2 to be enriched in gastric cancer prone to metastasis and proposed a potential biomarker function [43]. Recently, Fu et al. reported a similar impact of Tnni2 in pancreatic cancer [44]. Along with the orphan nuclear receptor ERRα, Tnni2 was identified as a crucial protein contributing to increased cell proliferation, migration and invasion of pancreatic cancer by up-regulation of Sirt1 and downstream Syt8 [44]. Within the acquired data ERRα or Syt8 could not be detected. However, Sirt1 was detected in TcnA treated HEp-2 cells with a log_2_ difference of −0.1, and a *p*-value of 0.07. TcnA treatment thus seems to cause a reduction of proliferation and migration which fits very well with the morphological changes and is also consistent with the effects of the other LCGTs. Modulating effects of TcnA on cellular proliferation would also be supported by the predicted kinase regulations (see below). As described by the examples of Lgmn and Tnni2 strong regulation of certain single proteins are not always easily explainable. However, regulation of pathways identified with IPA^TM^ seems to correlate well to known effects of TcnA.

Since TcnA inactivates Rho GTPases in a similar process to TcdA and TcdB of *C. difficile*, comparable downstream effects should be expectable [12]. Due to previous investigations from our laboratory on the cellular response after TcdA or TcdB treatment [21,22], we were able to compare the effects of TcnA with those of other LCGTs. On proteome level, activation of *Signaling by Rho Family GTPases*, *Actin Cytoskeleton Signaling*, *Epithelial Adherens Junction* and *Regulation of Actin-based Motility by Rho* were expectedly observed. In contrast, *RHOGDI Signaling* was inactivated (Appendix A). An overlap of 23 regulated pathways with both *C. difficile* toxins TcdA and TcdB was identified (Table 1). However, predicted pathway regulation between TcdB and TcnA was closer to each other than to TcdA regulations. Overall TcdA was more prone to inactivation of pathways, while many activated pathways were observed for TcnA and TcdB treatment (Figure 4). When having a deeper look into the pathway analysis (Appendix A) various proteins are assigned in different pathways, such as RhoA which obviously takes part in *RhoA signaling*, but also in *ILK* and *Integrin Signaling*, *Regulation of Actin-based Motility by Rho*, *RhoGDI Signaling* and of course *Signaling by Rho Family GTPases*. In order to draw reliable conclusions, not only the activation status of pathways but also the individual regulation of participating proteins is important, since a single pathway can also contribute to various cellular outcomes (data not shown). In order to identify these outcomes in more detail, functional analysis needs to be performed. Looking at the top-regulated single proteins there was a stronger correlation among *C. difficile* toxins instead of TcnA compared to TcdA/TcdB. TcdA and TcdB share RhoB and Heme oxygenase-1 (Hmox1) as some of the strongest up-regulated proteins. RhoB is a direct target of inactivation by *C. difficile* toxins and up-regulation of RhoB followed by toxin administration is a commonly observed effect of TcdA and TcdB [21,22,45,46]. Hmox1 however has not yet been discussed in the course of *C. difficile* toxin administration. It is a heat shock protein and catalyzes the conversion of heme to biliverdin, ferrous ion, and carbon monoxide. Activation is mainly induced as a response to stress especially when cells are challenged with oxidative stress which would go along with the observed increase of reactive oxygen species (ROS) production especially after TcdB administration [47,48,49,50]. Additionally, cytoprotective effects by preventing TNF-α induced apoptosis are mediated by Hmox1 [51]. In TcnA treated cells, only Hmox2 could be detected, but exhibited insignificant regulation. This goes along with the statement that Hmox1 and 2 catalyze the same reaction, however Hmox1 is inducible, while Hmox2 is constitutively expressed [52]. Hmox1 has been identified as a promising target for inflammatory bowel disease treatment due to its modulating effects on the immune response. This causes reduction of inflammation in the gut in a variety of animal models and promotes bacterial clearance by interfering in the bacterial electron transport chain [53,54,55]. Since no observation of Hmox1 regulation could be observed for TcnA treated cells it is tempting to postulate that TcnA treatment is not accompanied by excessively increased ROS production. This hypothesis would also be supported by superoxide dismutase [Cu-Zn] (SOD1), catalase (CAT) and thioredoxin (TXN) levels observed in the proteome. All of those proteins participate in cellular oxidative stress defense [56,57,58]. However, only SOD1 was observed as significantly regulated with a log_2_ difference of only 0.18 while the other two proteins were only slightly down-regulated and did not show significance.

Another eye-catching protein family being down regulated by treatment with LCGTs are tropomyosins (Tpm1-3, Table 2). Apart from the same isoform of Tpm2 being strongly down regulated in both *C. difficile* toxin treatments, TcdB treatment leads to down regulation of Tpm2 as well as Tpm3. In TcdA treatment, Tpm1 was identified as one of the top-3 down-regulated proteins. All three tropomyosins share binding to actin filament organization as a biological process [59]. This activity is present in muscle cells but is also observed in non-muscle cells. With Tpm’s binding to actin-filaments the cellular actin-cytoskeleton is stabilized. Vice versa down regulation of Tpms could be regarded as a proteomic marker for actin-cytoskeleton reorganization. In TcnA treatment seven isoforms of Tpm1, Tpm2 and Tpm4 were measured. Of those proteins six were significantly altered between TcnA treated cells and control cells, but regulation factors were below the threshold of factor 2 and only one isoform of Tpm2 showed a minimal down regulation. However, for Tpm1, interaction with the troponin complex has been reported and Tnni2 as mentioned above is one of the strongest down-regulated proteins in TcnA treatment. Putatively, tropomyosin regulation might occur in shorter incubation time and thus, the chosen 24 h proteomic snapshot might be too long to detect stronger alterations.

In contrast to the proteome, more pronounced effects could be observed on phosphoproteome level. In every replicate 9427 phosphosites could be identified and quantified underlining the high reproducibility of the analysis. The regulatory effects of TcnA became particularly evident in the 230 responsive phosphosites which were characterized by a significant *p*-value (*p* < 0.05) and strongly altered abundance (regulation factor > 2; <−2). Some of the strongest regulated phosphosites will be discussed hereinafter.

Serine/threonine-protein kinase N2 (Pkn2) has been identified as a protein worth discussing due to its regulated phosphosite Thr-124. TcnA treatment led to significantly (−log_10_
*p*-value: 3) increased phosphorylation at this site with a log_2_ regulation factor of 2.7 (Figure 5). Pkn2 is a Rho/Rac effector protein which takes part in various cellular processes such as signal transduction, cell cycle progression, actin cytoskeleton assembly, cell adhesion and others [60,61,62,63]. Pkn2 is known for binding RhoA, subsequently promoting a variety of RhoA-regulated processes. The regulated phosphosite at position 124 is located within the binding domain of RhoA [60]. Phosphorylation at this site leads to increased binding of Pkn2 to RhoA, thus promoting RhoA signaling. Degradation of RhoA via Smurf1 is hampered when Pkn2 is bound [60], which would go along with the observed slight increase in RhoA protein abundance (log_2_ difference of 0.36), and the observed activation of RhoA signaling pathway. What remains unclear was the effect of the N-acetylglucosamination of RhoA, which would result in an inactivation of RhoA downstream effects. Additional analysis would be helpful to identify whether up-regulation of RhoA could be another side effect of TcnA treatment similar to the inactivation and up-regulation of RhoB in TcdA treatment [45,46].

Another Protein, which possessed a phosphosite sensible to TcnA treatment, was Plakophilin 3 (Pkp3). The phosphosite Ser-313 of Pkp3 has been identified as significantly (−log_10_
*p*-value: 5) down regulated with a log_2_ regulation factor of −2.9 (Figure 5). Pkp3 is localized in desmosomes, structures at the cell membrane that are extracellularly connected to cadherins of surrounding cells and intracellularly bound to intermediate filaments [64,65]. Ser-313 phosphosite has been identified as responsible for a decreased membrane localization of Pkp3 leading to reduced tricellular contacts [66]. The observed decrease in phosphorylation after TcnA treatment should lead to increased cell–cell adhesion, which contradicts the observed phenotype of cell rounding accompanied by loss of cell–cell contacts. However, in former analysis a loss of Pkp3 did not exhibit pronounced adhesion defects, because plakophilin 1 (Pkp1), another protein of the plakophilin family, can rescue Pkp3 knockout concerning adherens junctions [67]. Pkp1 could not be detected in any of our treatment conditions. Aside from its functions concerning organization of the cytoskeleton, additional effects within cellular stress response or control of protein biosynthesis are known, although not associated with the regulated phosphosite [68,69].

The focal adhesion adapter protein Paxillin is another protein that was identified to have TcnA responsive phosphosites. In TcnA measurements four phosphosites of Paxillin showed downregulation, the strongest at Serine-258 with a log_2_ difference of −2.2 (Appendix A). Additionally, downregulation of the Paxillin pathway based on phosphorylation data could be identified (Figure 5). Paxillin serves as a scaffold protein in reactions mainly associated with cell movement and migration depending on its phosphorylation status on specific Tyrosine and Serine residues [70]. Anti-apoptotic effects of Paxillin, depending on its phosphorylation status on various phosphosites, have been reported as well [71]. Interestingly, Paxillin dephosphorylation has been reported to be sensitive to treatment with other LCGTs, namely TcdA and TcsL, as well [14,72]. In TcdA experiments, Paxillin dephosphorylation could be identified at the Tyrosine in position 118 [72,73]. This phosphosite is primarily regulated by the tyrosine-protein kinase Src [74]. TcdA is able to directly bind to the catalytic Src domain, thereby reducing the phosphorylation of Tyrosine 118 of Paxillin independently of Rho glucosylation [72]. After mutating the Tyrosine at position 118 to Phenylalanine the anti-apoptotic effects of Paxillin were suppressed [71]. TcsL dephosphorylation of Paxillin does not result from inactivation of Src but has been contributed to earlier cellular response events [14]. The phosphosite Ser-258 which was identified as down-regulated within our experiment has been identified to be regulated by Pak1 [75]. Against the background of HIV infection, its phosphorylation is associated with inhibition of the TNFα converting enzyme ADAM 17 binding to Paxillin, leading to an inhibition of lipid raft transfer [75]. It remains to be elucidated whether TcsL and TcnA treated cells address dephosphorylation of Paxillin in a similar manner or if there is yet another possibility for LCGTs to target Paxillin signaling.

Finally, based on phosphosite regulation predictions about the activity of associated kinases are possible. In our analysis activation for ZIP kinase, CaMK II & IV and Chk1 kinase was predicted. Shared downstream effects result in alterations concerning cell cycle progression (CaMK IV & II, Chk1), apoptosis (Zip, CaMK II, Chk1), as well as cell morphology (CaMK II &IV) [76,77,78,79,80]. On the other hand, Amp-activated protein Kinase has been predicted as down-regulated based on substrate regulation. It is a key protein in keeping cellular ATP homeostasis and, upon its activation, especially energy-consuming processes such as CDK-mediated cell-cycle progression are suppressed [81,82,83]. However, there have to be additional regulatory effects surpassing the effect of Amp-activated protein kinase for CDKs after TcnA treatment since substrates expressing the general CDK kinase substrate motif tended to be down-regulated as well as CDK1 and CDK5 substrates, respectively. A possible explanation would be the regulation of CDKs via cyclins, regulatory proteins such as p38 or inhibiting enzymes such as Wee1 or Myt1 [84,85]. Aurora-A kinase was the kinase with strongest inactivation predicted and has already been observed as inhibited after treatment of Hela cells with Toxin B from *C. difficile* serotype F strain 1470 (TcdBF) [86]. This takes part in cell proliferation and is usually activated during G2 phase to M phase transition. Degradation of Aurora A kinase has been related to enhanced apoptosis while its up-regulation is frequently observed in breast cancer [87]. All in all, kinase prediction leads to the conclusion that TcnA treatment decreases cellular proliferation in a complex pattern.

In this study we started with establishing a well-suited control concerning the analysis of TcnA effects on HEp-2 cells by using inactivated FA_TcnA. We were able to observe distinct regulations of TcnA on the proteome and phosphoproteome of HEp-2 cells, which could be illustrated not only via direct regulation of single proteins or phosphosites but also by observing differentially affected pathways on phospho- or protein level. Various pathways seemed to be affected similarly to TcdB-treated HEp-2 cells such as *Integrin Signaling* and *Paxillin Signaling* while other pathways such as *Actin Cytoskeleton Signaling* or *Signaling by Rho Family GTPases* showed activation across treatments with TcnA, TcdA and TcdB. A similar impact on the cellular proteome in comparison to TcdB would also fit to the close substrate GTPase profiles of TcdB and TcnA [73]. With the phosphosite regulation of Paxillin, a common target protein not only of TcnA and TcdA but also TcsL was identified. It might be promising to have a closer look, especially concerning the time scale when these phosphorylation events take place in order to match the identified regulations with cellular functions. Interestingly, compared with TcdA and B, there has been no evidence of increased ROS production by TcnA under the chosen conditions. All in all, a broad first insight into (phospho-) proteomic regulations of TcnA treated HEp-2 cells could be illustrated. However, remodeling time scaling and experimental set up should enable conclusions that are even more precise and, of course, functional analysis of regulated phosphosites/proteins would help break down the detailed effects caused in target cells. Apart from this compartmental enrichment for e.g., the case of legumain would be helpful in order to identify the functional effects underlying its regulation status.

## 4. Materials and Methods

### 4.1. Purification and Inactivation of Clostridium Novyi α-Toxin

Purification of α-Toxin (TcnA) from *Clostridium novyi* strain ATCC 19,402 was performed as previously described [88]. Briefly, the bacteria were cultivated (BHI, Difco, Sparks, MD, USA) in a dialysis bag for 72 h at 37 °C under microaerophilic conditions. Culture supernatant was ammonium sulfate precipitated and the protein pellet was extracted with buffer A (20 mM Tris, 15 mM NaCl pH 8.0) and dialyzed against the same buffer. This fraction was separated by anion exchange chromatoghraphy (HiPrep^TM^ Q Fast Flow 16/10 column, Sigma-Aldrich, St. Louis, MO, USA). Proteins were eluted by a linear gradient from 15 mM NaCl to 1 M NaCl in buffer A. Fractions were tested for their toxicity on MEF-cells. Fractions that showed rounding of cells after 4 h of incubation were combined and concentrated with Vivaspin 2 centrifugal concentrators (100 MWCO, Sigma-Aldrich). Seventy-five percent Glycerin was added in a ratio of 1:2 (*v*/*v*) for stabilization purposes. Aliquots were stored at −80 °C. Protein content of the TcnA concentrate was calculated using DC protein assay (Bio-Rad, Hercules, CA, USA). For control experiments, part of the purified TcnA was inactivated by incubation with 3.7% formaldehyde (Sigma-Aldrich, MO, USA) at 37 °C for 3 h. Inactivated toxin was referred to as FA_TcnA. Activity of TcnA was evaluated by cell rounding of HEp-2 cells using phase-contrast microscopy.

### 4.2. Cell Culture

Cell Culture was proceeded as previously described [89]. HEp-2 cells were maintained in a 75 cm^2^ flask in humidified atmosphere at 37 °C and 5% CO_2_. Cells were cultured in minimal essential medium (Gibco^TM^, Paisley, UK) supplemented with 10% fetal bovine serum, 100 U/mL penicillin and streptomycin each (Sigma-Aldrich, St. Louis, MO, USA). Cells were split, depending on confluency, in order to maintain vitality.

### 4.3. Toxin Treatment of HEp-2 Cells and Sample Preparation

At a confluency of 75% medium was replaced with 20 mL of fresh medium. For treatment 255 ng/mL of TcnA or FA_TcnA were added to 20 mL medium. Control cells were grown in medium without addition of toxin. Morphological changes of cells were documented by phase-contrast microscopy after 24 h. These conditions were chosen in order to reach maximal percentage of rounded cells without any undesired side effects as necrosis. Cells were washed twice with ice-cold PBS and stored at −80 °C overnight. The cells were then lysed by scraping them in 300 μL lysis buffer (8 M Urea, 50 mM ammonium bicarbonate (pH 8.0), 1 mM sodium ortho-vanadate, complete EDTA-free protease inhibitor cocktail (Roche, Mannheim, Germany), and phosphoSTOP phosphatase inhibitor cocktail (Roche, Mannheim, Germany)). Lysates were sonicated on ice for 15 s at 30% energy output. Cell debris was removed by centrifugation for 15 min at 13,000× *g* at 4 °C. The supernatant was used directly or stored at −80 °C.

### 4.4. Protein Digestion

Tryptic digestion of proteins proceeded as previously described [89]. Protein concentrations were calculated using DC protein assay (Bio-Rad, Hercules, CA, USA). Proteins were reduced with 5 mM DTT at 37 °C for 1 h and afterwards alkylated with 10 mM iodoacetamide at RT for 30 min. Alkylation was quenched by adding DTT to a final concentration of 5 mM. For digestion, lysates were diluted 1:5 with 50 mM ammonium bicarbonate (ABC) buffer. One mg protein was digested per condition at 37 °C for 4 h using 10 μg Lys-C (Wako, Osaka, Japan) followed by overnight digestion with 10 μg trypsin (Promega, Madison, WI, USA). Digestion was stopped by adding TFA to a final concentration of 1%. Peptide solutions were desalted with Sep Pak C18 1cc cartridges (Waters Corporation, Milford, MA, USA).

### 4.5. Tandem Mass Tags Labeling and Phosphopeptide Enrichment

From each sample 1 mg peptides were labeled with TMT10plex™ Isobaric Label Reagent (Thermo Scientific^TM^, Rockford, IL, USA) according to the manufacturer’s protocol. Labeled samples were pooled and 1 mg equivalent was used for peptide measurements (proteome data) while 5 mg equivalent was used for phosphopeptide enrichment. Both samples were again desalted with Sep Pak C18 1cc cartridges (Waters Corporation, Milford, MA, USA).

Phosphopeptides were enriched using High-Select™ TiO2 Phosphopeptide Enrichment Kit (Thermo Scientific^TM^, Rockford, IL, USA) according to the manufacturer’s protocol. All flow-throughs after application of the sample were collected and dried by vacuum centrifugation. The residues were then applied to High-Select™ Fe-NTA Phosphopeptide Enrichment Kit (Thermo Scientific^TM^, Rockford, IL, USA) enabling reproducible enrichment of phosphopeptides with a higher degree of phosphorylation. Eluates of both enrichment procedures (TiO2; Fe-NTA) were pooled and eight fractions generated by high pH reversed-phase peptide fractionation (Thermo Scientific^TM^, Rockford, IL, USA).

### 4.6. LC-MS Analysis

Dried peptides were dissolved in 0.1% TFA/2% ACN and analyzed in an Orbitrap Fusion Lumos mass spectrometer (Thermo Fisher Scientific) equipped with a nano-electrospray source and connected to an Ultimate 3000 RSLC nanoflow system (Thermo Fisher Scientific). Peptides were loaded onto a C18 PepMap100 (5 µm, 100 Å) µ-Precolumn (Thermo Fisher Scientific) and separated by a 50 cm µPAC^TM^ analytical column (Pharma fluidics) at 35 °C column temperature. A binary gradient with solvent A consisting of 0.1% formic acid (Merck, Darmstadt, Germany) and solvent B consisting of 100% ACN/0.1% formic acid was used to separate eluting peptides. Columns were equilibrated with 3.4% solvent B for 5 min. Separation of peptides followed by increasing B to 21% within 60 min and up to 42% in the following 32 min. Within 2 min B was increased to 75.6% which was held for 3 min and then decreased back to 3.4%. Re-equilibration of the column was reached by keeping B at 3.4% for 16 min. The spray voltage was set to 2 kV. Measurements were done with a DDA approach, with a cycle time of 3 s and Top N setting. Dynamic exclusion was set to 60 s, AGC target at 4 × 10^5^ and a maximum injection time of 50 ms was applied. Orbitrap resolution for the MS1 scan was set to 120,000 and higher-energy collisional dissociation (HCD) fragmentation at 38% was used followed by MS2 measurements. The first mass was set to 100 *m*/*z* in order to be able to quantify the split reporter ions ranging from 126–131 *m*/*z*. MS2 maximum injection time was set to 110 ms and corresponding isolation width to 0.8 *m*/*z*. Orbitrap resolution for MS2 scans was set to 60,000.

### 4.7. Data Processing

Raw data were processed with MaxQuant software (version 1.6.14.0, Martinsried, Germany) [90] using the Andromeda search engine [91]. Spectra were searched against the Swiss-Prot reviewed UniprotKB *Homo sapiens* database (version 01/2020, 42,352 entries) [92]. Carbamidomethylation of cysteine was set as fixed modification while oxidation of methionine, N-terminal acetylation, deamidation of glutamine and asparagine were set as variable modifications. N-acetylglucosaminylation (C_8_H_13_NO_5_; 203.079 Da) was considered as a variable modification on Threonine residues due to TcnA treatment. For Phosphopeptide-enriched samples the variable modification phosphorylation at serine, threonine, and tyrosine residues was considered as well. Reporter ion MS2 for TMT 10-plex was set with the associated isotopic distributions. Second peptide search was activated and match between runs (0.7 min match time window; 20 min alignment time window) enabled. False discovery rate was set to 0.01 and a maximum of two missed cleavages was allowed.

Only proteins that were quantified in every replicate were used for further data processing. Phosphosites additionally needed to have a localization probability of at least 75%. Both proteome and phosphoproteome were normalized by subtracting the mean intensity of each sample. Phosphosite analysis was solely performed if the corresponding protein could be quantified within the proteome measurements. Phosphosite abundancies were normalized to the corresponding protein abundancy. Thus, identified differences trace back to the degree of phosphorylation instead of protein expression.

Data evaluation and analysis were done using Perseus (1.6.14.0) [93] and Microsoft Excel (Microsoft Corporation, 2018. Microsoft Excel, Available online: https://office.microsoft.com/excel, 26 August 2022). Data visualization was done with GraphPad Prism (GraphPad Prism 7.00 version for Windows, GraphPad Software, San Diego, CA, USA, www.graphpad.com). Upstream and canonical pathway analyses were performed using ingenuity pathway analysis (IPA; QIAGEN Inc., Hilden, Germany, https://digitalinsights.qiagen.com/IPA, 26 August 2022) [94]. This software predicts regulatory proteins by comparing up- and down-regulated proteins (or phosphosites) to a manually curated database. Even if a protein could not be identified itself, conclusions can be made due to known relations, e.g., based on phosphorylation data predictions about the activity of certain kinases can be made. For IPA analysis all significantly different phosphosites and proteins were used (*p*-value > 0.05).

## Figures and Tables

**Figure 1 ijms-23-09939-f001:**
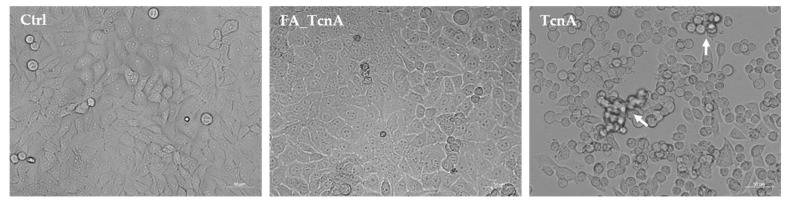
Morphological alterations of HEp-2 cells after 24 h of incubation with TcnA (Magnification 100). Ctrl: untreated cells, FA_TcnA: cells treated with inactivated TcnA 255 ng/mL, TcnA: cells treated with 255 ng/mL TcnA. White arrows indicating cell strands ranging into the medium but still attached to cells adhering to the flask.

**Figure 2 ijms-23-09939-f002:**
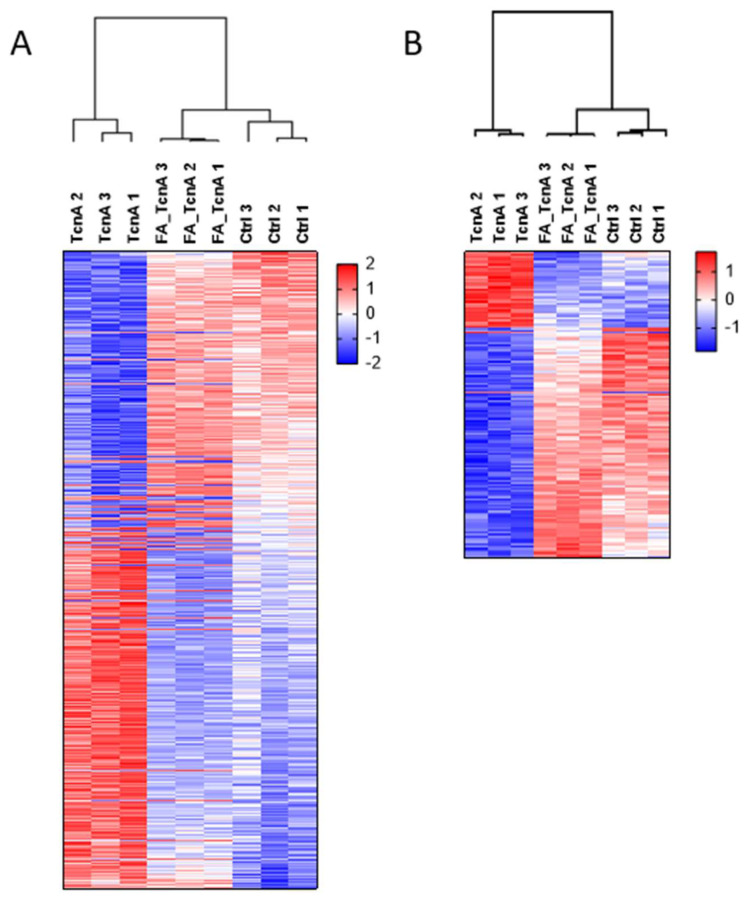
Statistical analyses and quality control of effects of TcnA on the proteome of HEp-2 cells. (**A**) Heat map of significantly changed proteins based on BJH-corrected and FDR-based ANOVA testing of the protein abundancies. (**B**) Heat map of significantly changed phosphosites based on BJH-corrected and FDR-based ANOVA testing of the protein abundancies. Dendrograms show similarity of samples/treatments.

**Figure 3 ijms-23-09939-f003:**
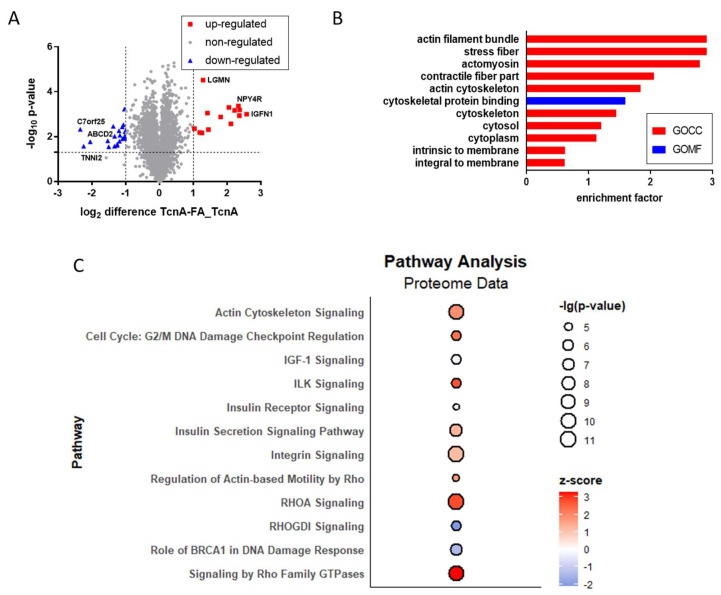
Effects of TcnA on the proteome of HEp-2 cells. (**A**) Volcano plot of proteins from HEp-2 cells with TcnA and FA_TcnA. Dashed lines give the minimal regulation factor (log_2_ difference > 1; <−1) and the significance level −log_10_ (0.05). (**B**) Fisher’s exact test of significantly regulated proteins after BJH corrected *t*-testing. Enrichment factor > 1 indicates enrichment of the term, while an enrichment factor < 1 marks a reduction of this term within the significant proteins. (**C**) IPA^TM^ Pathway analysis of significant proteins (*p*-value < 0.05). A positive z-scores indicates predicted activation of the pathway, while a negative z-score indicates predicted inhibition of the pathway. Size of the points correlates to the −log_10_ (*p*-value).

**Figure 4 ijms-23-09939-f004:**
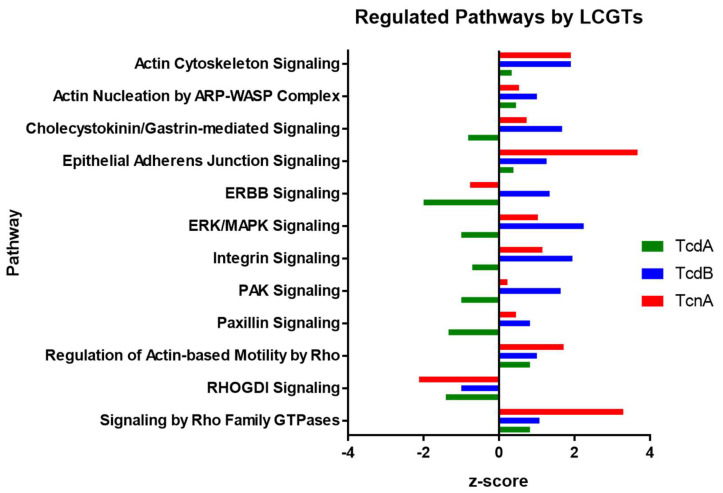
Regulated pathways (*p*-value < 0.05) in TcdA [21], TcdB [22] and TcnA treated HEp-2 cells. Positive z-score indicates activation (z-score > 2 significant activation), while a negative z-score indicates inhibition of the pathway (z-score < −2 significant inhibition).

**Figure 5 ijms-23-09939-f005:**
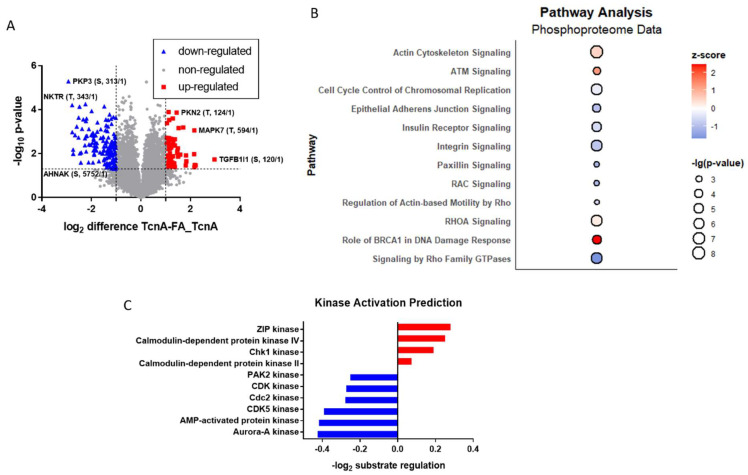
Effects of TcnA on the phosphoproteome of HEp-2 cells. (**A**) Volcano plot of proteins from HEp-2 cells with TcnA and FA_TcnA. Dashed lines give the minimal regulation factor (log_2_ difference > 1; <−1) and the significance level −log_10_ (0.05). (**B**) IPA^TM^ Pathway analysis of significantly altered phosphosites (*p*-value < 0.05). Size of the points correlates to the −log_10_ (*p*-value). (**C**) Kinase activation prediction. Top 10 regulated kinases identified by Fisher’s exact test (*p*-value < 0.05) and activation status prediction based on phosphosite (*p* < 0.05) regulation.

**Table 1 ijms-23-09939-t001:** Regulated pathways (*p* < 0.05) identified in TcdA, TcdB and TcnA treated HEp-2 cells. Pathways sorted according to regulation following TcnA treatment.

Pathway	z-Score		
	TcdA	TcdB	TcnA
Signaling by Rho Family GTPases	0.82	1.07	3.29
Epithelial Adherens Junction Signaling	0.38	1.26	3.67
IL-8 Signaling	0.82	2.11	2.12
Protein Kinase A Signaling	0.82	1.9	2.11
Actin Cytoskeleton Signaling	0.33	1.9	1.9
Regulation of Actin-based Motility by Rho	0.82	1	1.71
Ephrin Receptor Signaling	−1.13	1.67	1.46
Leukocyte Extravasation Signaling	−1.13	1.67	1.3
mTOR Signaling	0.45	1.41	1.22
Integrin Signaling	−0.71	1.94	1.15
Hepatic Fibrosis Signaling Pathway	−0.63	1.29	1.07
ERK/MAPK Signaling	−1	2.24	1.03
Cholecystokinin/Gastrin-mediated Signaling	−0.82	1.67	0.73
Cardiac Hypertrophy Signaling	0.82	1.67	0.67
Actin Nucleation by ARP-WASP Complex	0.45	1	0.53
Paxillin Signaling	−1.34	0.82	0.45
PAK Signaling	−1	1.63	0.22
HER-2 Signaling in Breast Cancer	−1	1.63	−0.18
Glioma Invasiveness Signaling	0.45	0.82	−0.24
HGF Signaling	−1.34	1.41	−0.24
ERBB Signaling	−2	1.34	−0.77
HIPPO signaling	−0.82	−1	−1.15
RHOGDI Signaling	−1.41	−1	−2.12

**Table 2 ijms-23-09939-t002:** Top 10 up- and down-regulated proteins after TcdA, TcdB and TcnA treatment. Bold: proteins that are regulated in more than one treatment condition.

Experiment	Up-Regulated				Down-Regulated			
	Gene Name	Protein ID	*p*-Value	Log_2_ Fold-Change	Gene Name	Protein ID	*p*-Value	Log_2_ Fold-Change
TcnA	IGFN1	Q86VF2-5	9.9 × 10^−4^	2.58	C7orf25	Q9BPX7	4.8 × 10^−3^	−2.34
TcnA	NKTR	P30414	6.4 × 10^−4^	2.37	TNNI2	P48788-2	2.7 × 10^−2^	−2.24
TcnA	FBRSL1	Q9HCM7	1.2 × 10^−3^	2.36	ABCD2	Q9UBJ2	1.7 × 10^−2^	−2.05
TcnA	NPY4R	P0DQD5	4.3 × 10^−4^	2.33	SMU1	Q2TAY7-2	1.5 × 10^−2^	−1.53
TcnA	INTS1	Q8N201	6.8 × 10^−4^	2.22	C1QTNF6	Q9BXI9-1	2.8 × 10^−2^	−1.50
TcnA	MTCL1	Q9Y4B5-2	2.7 × 10^−3^	2.11	MIF	P14174	3.5 × 10^−3^	−1.37
TcnA	TAOK1	Q7L7X3	5.0 × 10^−4^	2.05	CCDC129	Q6ZRS4-2	2.7 × 10^−2^	−1.33
TcnA	MYLPF	Q96A32	1.3 × 10^−3^	1.81	SCAF4	O95104-2	9.7 × 10^−3^	−1.33
TcnA	C12orf29	Q8N999-3	4.8 × 10^−3^	1.45	NUDT9	Q9BW91-2	2.5 × 10^−2^	−1.25
TcnA	SLC6A12	P48065	8.8 × 10^−4^	1.42	HSP90AB2P	Q58FF8	2.2 × 10^−2^	−1.25
TcdB	**RHOB**	**P62745**	**1.4 × 10^−2^**	**2.25**	CDC42EP1	Q00587	2.1 × 10^−3^	−2.48
TcdB	CTSL	P07711	1.1 × 10^−4^	1.94	RND3	P61587	4.5 × 10^−3^	−1.63
TcdB	JUN	P05412	5.6 × 10^−4^	1.73	**YAP1**	**P46937**	**7.6 × 10^−6^**	**−1.55**
TcdB	**HMOX1**	**P09601**	**2.0 × 10^−2^**	**0.83**	CDC42EP2	O14613	1.2 × 10^−2^	−1.30
TcdB	MYO10	Q9HD67	1.8 × 10^−2^	0.77	CYR61	O00622	7.2 × 10^−3^	−1.23
TcdB	SDCBP	O00560	3.2 × 10^−4^	0.77	TPM2	Q5TCU3	1.3 × 10^−2^	−1.05
TcdB	SEC14L1	Q92503	6.7 × 10^−3^	0.76	**TPM2**	**P07951-2**	**1.6 × 10^−2^**	**−0.86**
TcdB	SDCBP	O00560-2	2.2 × 10^−2^	0.71	TPM3	Q5VU63	8.1 × 10^−3^	−0.74
TcdB	STAU1	O95793	2.5 × 10^−2^	0.63	F3	P13726	3.3 × 10^−2^	−0.69
TcdB	RASSF5	Q8WWW0	3.3 × 10^−2^	0.61	SLC1A3	P43003	1.9 × 10^−2^	−0.67
TcdA	**RHOB**	**P62745**	**1.8 × 10^−4^**	**2.72**	**YAP1**	**P46937**	**4.5 × 10^−4^**	**−1.71**
TcdA	**HMOX1**	**P09601**	**4.4 × 10^−2^**	**0.93**	**TPM2**	**P07951-2**	**1.2 × 10^−5^**	**−1.14**
TcdA	APP	E9PEV0	1.0 × 10^−3^	0.67	TPM1	P09493-3	7.6 × 10^−4^	−1.07
TcdA	TIMP3	B1AJV7	3.4 × 10^−2^	0.61	CLDN1	O95832	5.2 × 10^−3^	−0.78
TcdA	TM4SF1	F8WF27	2.5 × 10^−3^	0.61	SLC1A3	P43003-2	3.6 × 10^−2^	−0.74
TcdA	TUBB2A	Q13885	3.3 × 10^−2^	0.55	AFAP1	Q8N556	2.6 × 10^−3^	−0.73
TcdA	CD63	F8VNT9	6.7 × 10^−3^	0.51	FHL2	J3KNW4	2.1 × 10^−3^	−0.63
TcdA	RRM2	P31350	7.2 × 10^−3^	0.51	MPRIP	Q6WCQ1	1.9 × 10^−4^	−0.51
TcdA	S100A16	Q96FQ6	1.4 × 10^−3^	0.49	STAT3	K7ENL3	4.2 × 10^−3^	−0.46
TcdA	ITGA5	P08648	5.0 × 10^−3^	0.46	PRKCI	P41743	3.1 × 10^−2^	−0.46

## Data Availability

The mass spectrometry proteomics data have been deposited to the ProteomeXchange Consortium via the PRIDE [95] partner repository with the dataset identifier PXD036078 and 10.6019/PXD036078.

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
