# Peer review of "Clostridium**n**ovyi’s* Alpha-Toxin Changes Proteome and Phosphoproteome of HEp-2 Cells"

_ijms, 2022, doi:10.3390/ijms23179939_

Round 1
Reviewer 1 Report
The authors of the paper “Clostridium novyi's Alpha-Toxin changes proteome and phosphoproteome of HEp-2 cells” developed a good job describing the effect of Alpha-toxin A (TcnA) on HEp-2 cells at the proteomic level, which is of great interest. However, I should make several comments:
1-. I can't find the access code to the proteomics data in a public repository. This, today, is essential in proteomics studies. Please, indicate the code so that the results can be reviewed. If the data has not been uploaded, please upload it.
2. The authors have not performed independent validations to experimentally verify changes in expression levels of differentially expressed proteins or changes in activity of differentially phosphorylated proteins. This, or any other functional validation that supported the omics data, is required.
3. I don't know if it's because of the version of the manuscript that I downloaded (I don't think so, since it's in PDF) or the original file from the authors, but many of the references in the text are wrong. Where the references should be there are error messages. Please, fix.
4. Why did the authors use 225 ng/mL of TcnA? Please, explain.
5. Although in figure 1 (which is actually figure 2, please, fix) it can be seen that 3 replicates were used for each condition, this data should also be included in the materials and methods.
6. The font in figure 1 is different from the rest of the text. Please, fix.
7. Please, remove the frame to panel C in figure 3.
8. Table 1 is cut into two pages. Please, fix.
9. The authors compare, in the results section, the data obtained by TMT in treatment with TcnA with those obtained, in previous works, by SILAC with TcdA and TcdB. Beyond the difference in the methodology used, other factors such as the batch effect can influence the results. Therefore, I would encourage the authors to indicate, for this work in the results section, to present the 35 differentially expressed proteins (14 up-regulated and 21 down-regulated, lines 139-148), leaving the information contained in the table 2 for the discussion section.
10. The authors carry out phosphosite analysis only if the corresponding protein could be quantified. However, they are two different levels of regulation, so that the phosphorylation of a protein can completely change its activity profile without changing its expression profile. However, I understand that it is a parameter established by the authors as a sieve, and that they have relied on it, but I would recommend that it be discussed in the corresponding section.
Author Response
Comments to the Reviewers
Schweitzer et al.
“Clostridium novyi’s alpha-Toxin Changes Proteome and Phosphoproteome of HEp-2 Cells”.
Response to Reviewer 1
The authors of the paper “Clostridium novyi's Alpha-Toxin changes proteome and phosphoproteome of HEp-2 cells” developed a good job describing the effect of Alpha-toxin A (TcnA) on HEp-2 cells at the proteomic level, which is of great interest. However, I should make several comments:
We thank reviewer 1 for these encouraging words.
1-. I can't find the access code to the proteomics data in a public repository. This, today, is essential in proteomics studies. Please, indicate the code so that the results can be reviewed. If the data has not been uploaded, please upload it.
We thank reviewer 1 for this comment. The MS data including all raw files have been uploaded to PRIDE DB at EMBL and we have just recently received the accession No: PXD036078 and 10.6019/PXD036078 that is now included in the manuscript.
- The authors have not performed independent validations to experimentally verify changes in expression levels of differentially expressed proteins or changes in activity of differentially phosphorylated proteins. This, or any other functional validation that supported the omics data, is required.
We thank reviewer 1 for this comment. Validation experiments can be carried out using Western blotting or flow cytometry in single or multiplexed way. For both methods suitable antibodies are necessary that have to detect alterations on proteome level with similar sensitivity. Particular for phospho site analysis such antibodies are rare. In the earlyr times (15 years ago) of MS-based proteomics such experiments have been carried out quite frequently, but unfortunately, their benefits were week. On the other hand, MS-based proteomics has developed to a sophisticated method producing reliable and comprehensive data sets which are only limited by sensitivity. For this point, we recommend a review from Aebersold and Mann published by Nature (Aebersold R, Mann M., Nature. 2016; 537:347-355. PMID: 27629641).
The presented results on the effects of TcnA on epithelial cells have been obtained after triplicate analysis and only statistical significant (p-value <0.05) alterations with an at least two-fold change in abundance were considered. Moreover, the results described herein were validated by results received from comparable experiments with toxins from C. difficile (Fig. 4 and Table 1, 2).
- I don't know if it's because of the version of the manuscript that I downloaded (I don't think so, since it's in PDF) or the original file from the authors, but many of the references in the text are wrong. Where the references should be there are error messages.
We thank reviewer 1 for this comment and apologize this point. We have removed some of the MS Word linkings and now it works.
- Why did the authors use 225 ng/mL of TcnA? Please, explain.
We thank reviewer 1 for this comment. 255ng/mL is the protein concentration calculated by DC protein assay of the toxin preparation. The chosen concentration was determined experimentally in order to reach comparable morphological changes after similar incubation times as observed in TcdA and TcdB experiments. We chose HEp-2 cells, a HeLa derived cell line, an often been exploited cell culture model in proteome research in the bacterial toxins field (PMC8477661 PMC6304397 PMID: 27860399). With regard to Tcna, HEp-2 seems to be a suitable cell culture model, as HEp-2 cell express LDLR and SLC35B2, factors in involved in the cellular uptake of TcnA. HEp-2 cells, however, were not as susceptible to Tcna as the related Toxin A and B from C. difficile. Therefore, relative to TcdA and TcdB higher Tcna concentrations were required for inducing biological effects.
- Although in figure 1 (which is actually figure 2, please, fix) it can be seen that 3 replicates were used for each condition, this data should also be included in the materials and methods.
We thank reviewer 1 for this comment and emended this point.
- The font in figure 1 is different from the rest of the text. Please, fix.
We thank reviewer 1 for this comment and emended this point.
- Please, remove the frame to panel C in figure 3.
We thank reviewer 1 for this comment and emended this point.
- Table 1 is cut into two pages. Please, fix.
We thank reviewer 1 for this comment and emended this point.
- The authors compare, in the results section, the data obtained by TMT in treatment with TcnA with those obtained, in previous works, by SILAC with TcdA and TcdB. Beyond the difference in the methodology used, other factors such as the batch effect can influence the results. Therefore, I would encourage the authors to indicate, for this work in the results section, to present the 35 differentially expressed proteins (14 up-regulated and 21 down-regulated, lines 139-148), leaving the information contained in the table 2 for the discussion section.
We thank reviewer 1 for this comment. The top ten regulated proteins in Table 2 were only identified in the present work. All other data of protein intensities are available in supplemental Table 1. We choose this option for layout reason and to keep the manuscript a little shorter.
- The authors carry out phosphosite analysis only if the corresponding protein could be quantified. However, they are two different levels of regulation, so that the phosphorylation of a protein can completely change its activity profile without changing its expression profile. However, I understand that it is a parameter established by the authors as a sieve, and that they have relied on it, but I would recommend that it be discussed in the corresponding section.
We thank reviewer 1 for this comment. Since we observed only weak alterations on the proteome level, it made sense to normalize phosphopeptide intensities to the proteom level. Differences in replicates were obvious if no normalization step was carried out and obtained intensity values differ clearly. This was probably caused by the methods necessary for phosphopeptide enrichement e.g. TiO2 chromatography.
Reviewer 2 Report
The manuscript by Theresa Schweitzer et al entitled "Clostridium novyi’s alpha-Toxin Changes Proteome and Phosphoproteome of HEp-2 Cells" presents a comprehensive proteomic and phospho-proteomic analyses of Hep-2 cells treated with TcnA in order to decipher signaling pathways possibly explaining TcnA-mediated apoptosis. MS-based data are analyzed and interpreted for pathway analyses.
Mayor comments:
1. What protein was used as "MS-standard" for protein level changes? Actin?
2 Stemming from Figure 3C, the discussion needs some aspects of signaling pathway overlap and branching in general and for this study/data interpretation.
3. Signaling pathway analyses stemming from MS-based proteomics is known to be a great start to build a molecular signaling model; however, only additional biochemical analyses to verify the findings and conclusions are required such as in this case immunoblot analyses. Please comment.
4. The discussion needs to have more in-depth interpretation of data stemming from proteomics vs. phospho-proteomics analyses. Regulation of protein levels is for example by protein synthesis (up-regulation) or protein degradation or protein release (down-regulation).Additionally, changes in phosphorylation levels is associated to base protein levels which would be linked to the comment above that phosphorylation levels of a given protein need to be analyzed in the context of its total protein level in a defined experimental condition.
5. Confirmation of signaling pathway analyses and models are often carried out in experiments using targeted inhibitors. In this study, key phosphoproteins could be regulated in order to verify interpretation. Please comment.
6. The discussion needs to comment on the sugar-modifications mentioned in the abstract vs. phosphorylation as they occur both on threonine site and its impact on the MS-data and signaling concept.
General comments:
Many commas are missing!
1. Nice introduction tailored to the topic!
2. l.72, please add reference to DDA proteomic analysis.
3. Somewhere in the manuscript it should be mentioned why Hep-2 cells were used for this study.
4. Please clarify the concentration of TcnA used as the M&M section mentioned 5.1ug while the results section states 255 ng/mL.
5. Figure 1 quality needs to be improved as very faint.
6. l.100: what is 'protein homeostasis'?
7. l.108-110: Please elaborate a on the BJH concept and provide a reference.
8. Heat map figure is mislabeled, would be Figure 2.
9. The conclusion of FA_TcnA being an appropriate control is nicely done!
10. l.142: the reference to 14 and 21 proteins is a bit misleading as 'only' three proteins each 'side' are labeled in Figure 3A. Maybe rephrase and adjust the figure legend accordingly.
11. Figure 3C needs more explanation. The -lg(p-value) is not clear. Would a p-value of 0.01 be +4.6?
12. The paragraph spanning l.172 to l.188 is very heavy on MS terminology. The difference between p-value and z-score is not easy to follow and would need some clarification. The statement of "...gives a hint of..." is scientifically not really convincing. Please explain in more detail and see general comment re confirmation of data interpretation.
13. The references [20] and [21] should be mentioned also in the figure legend of Figure 4.
14. Please move table 1 down to have it displayed on one page, Also modify the table legend that the pathways are sorted by increasing z-score of the TcnA-treated samples.
15. l.391: What is meant with 'shorter incubation time'? In case the TrnA incubation, then this should be justified in the M&M section why 24 hrs were chosen in this study.
16. l.426: Please fix the reference to [64].
Author Response
Comments to the Reviewers
Schweitzer et al.
“Clostridium novyi’s alpha-Toxin Changes Proteome and Phosphoproteome of HEp-2 Cells”.
Response to Reviewer 2
The manuscript by Theresa Schweitzer et al entitled "Clostridium novyi’s alpha-Toxin Changes Proteome and Phosphoproteome of HEp-2 Cells" presents a comprehensive proteomic and phospho-proteomic analyses of Hep-2 cells treated with TcnA in order to decipher signaling pathways possibly explaining TcnA-mediated apoptosis. MS-based data are analyzed and interpreted for pathway analyses.
We thank reviewer 2 for this comment.
Mayor comments:
- What protein was used as "MS-standard" for protein level changes? Actin?
We thank reviewer 2 for this question. In MS based proteomics using TMT-labelling, all proteins are quantified within the same run. Post-measurement intensities of all proteins of one samples are usually normalized to the total mean protein abundance of this sample. Thus, differences between samples and replicates caused by protein load or other means are balanced.
2 Stemming from Figure 3C, the discussion needs some aspects of signaling pathway overlap and branching in general and for this study/data interpretation.
We thank reviewer 2 for this comment and changed the text accordingly.
- Signaling pathway analyses stemming from MS-based proteomics is known to be a great start to build a molecular signaling model; however, only additional biochemical analyses to verify the findings and conclusions are required such as in this case immunoblot analyses. Please comment.
We thank reviewer 2 for this comment and completely agree with him. Further functional analyses and/or immunoblot analyses are necessary to narrow down biological processes and direct interactions that lead to the observed proteomic changes. However, the present study should only outline different parts of cellular responses, which are affected by TcnA treatment. Particular antibodies for specific phospho sites are rare and not available for many phosphorylated proteins. However, MS-based proteomics has developed to a sophisticated method producing reliable and comprehensive data sets which are only limited by sensitivity. For this point, we recommend a review from Aebersold and Mann published by Nature (Aebersold R, Mann M., Nature. 2016; 537:347-355. PMID: 27629641).
- The discussion needs to have more in-depth interpretation of data stemming from proteomics vs. phospho-proteomics analyses. Regulation of protein levels is for example by protein synthesis (up-regulation) or protein degradation or protein release (down-regulation).Additionally, changes in phosphorylation levels is associated to base protein levels which would be linked to the comment above that phosphorylation levels of a given protein need to be analyzed in the context of its total protein level in a defined experimental condition.
We thank reviewer 2 for this comment. However, it is not completely clear to us. All determined phosphoprotein intensities were normalized to their corresponding protein intensities. Thus, regulation of phosphosites as stated in the manuscript show the up or down-regulation of all indicated phosphosite´, because protein regulation is already included in this calculation.
- Confirmation of signaling pathway analyses and models are often carried out in experiments using targeted inhibitors. In this study, key phosphoproteins could be regulated in order to verify interpretation. Please comment.
We thank reviewer 2 for this comment and completely agree with him. I think this remark takes the same direction as point no. 3). This study is a screening in order to identify cellular regulations that are worth being looked at in more detail. Further analysis, whether it is concerning the presence of certain phosphosites and direct interactions or more detailed information on the degree of regulation on kinases is absolutely interesting in order to pinpoint exact biological functions. However, this will be done in future studies and is underway in our laboratory.
- The discussion needs to comment on the sugar-modifications mentioned in the abstract vs. phosphorylation as they occur both on threonine site and its impact on the MS-data and signaling concept.
We thank reviewer 2 for this comment and agree, that this is a highly interesting point. It has already been addressed by Hart et al (2011, https://pubmed.ncbi.nlm.nih.gov/21391816/) who analysed that o-GlcNACylation and phosphorylation can compete for the same site in proteins. However, a phosphorylation on Thr-37 in Rho or Thr35 in Rac/Cdc42 has not been described so far. In earlier experiments, we tried in our laboratory to identify such a phosphorylation but were not successful with this experiment. Also in our comprehensive phosphorylation analysis described in the present paper, we did not identify a phosphorylation at Thr-37 or Thr-35.
However, on the Uniprot data base it is indicated for GTPases CDC42 and HRas that putatively phosphorylation on the target threonine residue might occure. We will keep this in mind in future analysis.
General comments:
Many commas are missing!
We thank reviewer 2 for this comment and corrected the text accordingly.
- Nice introduction tailored to the topic!
We thank reviewer 2 for this comment!
- l.72, please add reference to DDA proteomic analysis.
We thank reviewer 2 for this comment and corrected the text accordingly.
- Somewhere in the manuscript it should be mentioned why Hep-2 cells were used for this study.
We thank reviewer 2 for this comment and corrected the text accordingly.
- Please clarify the concentration of TcnA used as the M&M section mentioned 5.1ug while the results section states 255 ng/mL.
We thank reviewer 2 for this comment. We adjusted the specification in the M&M section. Both specifications were correct, but now it is more comprehensible.
- Figure 1 quality needs to be improved as very faint.
We thank reviewer 2 for this comment and improved the figure accordingly.
- l.100: what is 'protein homeostasis'?
We thank reviewer 2 for this comment. In this paragraph protein homeostasis is used to describe the global cellular protein expression of healthy, growing cells, without any form of stress when all cellular functions are balanced. The base level in this experiment is the untreated control and the samples treated with FA_TcnA. By addition of TcnA this protein homeostasis gets unbalanced because the cells are responding to the toxin leading to up- and down-regulation of various proteins and in- activation of signaling pathways.
- l.108-110: Please elaborate a on the BJH concept and provide a reference.
We thank reviewer 2 for this comment. A reference has been provided for Benjamini-Hochberg based testing and this statistical means is routinely used to analyse proteome and phospo proteome data sets to minimize the number of false positives within multiple T-Tests by adjusting the significance level or the p-value. However, for us it is not clear what we should elaborate on the BJH concept.
- Heat map figure is mislabeled, would be Figure 2.
We thank reviewer 2 for this comment and improved the figure accordingly.
- The conclusion of FA_TcnA being an appropriate control is nicely done!
We thank reviewer 2 for this comment!
- l.142: the reference to 14 and 21 proteins is a bit misleading as 'only' three proteins each 'side' are labeled in Figure 3A. Maybe rephrase and adjust the figure legend accordingly.
We thank reviewer 2 for this comment. Annotating all Proteins within the volcano plot will get very chaotic and not annotating any proteins will reduce the informational value. Some of the proteins are mentioned in the discussion, which is why we would prefer to let only them stay annotated.
- Figure 3C needs more explanation. The -lg(p-value) is not clear. Would a p-value of 0.01 be +4.6?
We thank reviewer 2 for this comment. The term “-lg” is generally used for “-log10”. In the graphic program, it is not possible to use subscripts. Thus, we choose this expression. Annyhow, –log10 was added in the figure description for clarity.
- The paragraph spanning l.172 to l.188 is very heavy on MS terminology. The difference between p-value and z-score is not easy to follow and would need some clarification. The statement of "...gives a hint of..." is scientifically not really convincing. Please explain in more detail and see general comment re-confirmation of data interpretation.
We thank reviewer 2 for this comment. Some of the sentences were rephrased to make it clearer.
- The references [20] and [21] should be mentioned also in the figure legend of Figure 4.
We thank reviewer 2 for this comment and changed the figure legends accordingly.
- Please move table 1 down to have it displayed on one page, Also modify the table legend that the pathways are sorted by increasing z-score of the TcnA-treated samples.
We thank reviewer 2 for this comment and modified table legend and sorting.
- l.391: What is meant with 'shorter incubation time'? In case the TcnA incubation, then this should be justified in the M&M section why 24 hrs were chosen in this study.
We thank reviewer 2 for this comment and modified the text in MM accordingly.